# Circulating U13 Small Nucleolar RNA as a Potential Biomarker in Huntington’s Disease: A Pilot Study

**DOI:** 10.3390/ijms232012440

**Published:** 2022-10-18

**Authors:** Silvia Romano, Carmela Romano, Martina Peconi, Alessia Fiore, Gianmarco Bellucci, Emanuele Morena, Fernanda Troili, Virginia Cipollini, Viviana Annibali, Simona Giglio, Rosella Mechelli, Michela Ferraldeschi, Liana Veneziano, Elide Mantuano, Gabriele Sani, Andrea Vecchione, Renato Umeton, Franco Giubilei, Marco Salvetti, Rosa Maria Corbo, Daniela Scarabino, Giovanni Ristori

**Affiliations:** 1Department of Neurosciences, Mental Health and Sensory Organs, Sant’Andrea Hospital, Sapienza University of Rome, 00189 Rome, Italy; 2Department of Human Neurosciences, Sant’Andrea Hospital, Sapienza University of Rome, 00189 Rome, Italy; 3Department of Human Neurosciences, Policlinico Umberto I, Sapienza University of Rome, 00185 Rome, Italy; 4Department of Biology and Biotechnology, Sapienza University of Rome, 00185 Rome, Italy; 5Department of Experimental Medicine, Sapienza University of Rome, Policlinico Umberto I, 00185 Rome, Italy; mailto:; 6IRCCS San Raffaele, San Raffaele Roma Open University, 00166 Rome, Italy; 7Ospedale San Giovanni Battista, Associazione dei Cavalieri Italiani del Sovrano Militare Ordine di Malta (ACISMOM), 00148 Rome, Italy; 8CNR Institute of Translational Pharmacology, 00185 Rome, Italy; 9Department of Psychiatry, Fondazione Policlinico Universitario Agostino Gemelli IRCCS, 00168 Rome, Italy; 10Department of Neuroscience, Section of Psychiatry, Catholic University of the Sacred Heart, 00168 Rome, Italy; 11Surgical Pathology Units, Department of Clinical and Molecular Medicine, Sant’Andrea Hospital, Sapienza University of Rome, 00189 Rome, Italy; 12Department of Informatics and Analytics, Dana-Farber Cancer Institute, Boston, MA 02215, USA; 13Department of Biological Engineering, Department of Mechanical Engineering, Massachusetts Institute of Technology, Cambridge, MA 02139, USA; 14Harvard School of Public Health, Boston, MA 02115, USA; 15Weill Cornell Medicine, New York, NY 10065, USA; 16IRCCS Istituto Neurologico Mediterraneo (INM) Neuromed, 86077 Pozzilli, Italy; 17CNR Institute of Molecular Biology and Pathology, 00185 Rome, Italy; 18Neuroimmunology Unit, IRCCS Fondazione Santa Lucia, 00179 Rome, Italy

**Keywords:** fluid biomarkers, small circulating non-coding RNAs, small nucleolar RNAs, Huntington’s disease

## Abstract

Plasma small RNAs have been recently explored as biomarkers in Huntington’s disease (HD). We performed an exploratory study on nine HD patients, eight healthy subjects (HS), and five psychiatric patients (PP; to control for iatrogenic confounder effects) through an Affymetrix-Gene-Chip-miRNA-Array. We validated the results in an independent population of 23 HD, 15 pre-HD, 24 PP, 28 Alzheimer’s disease (AD) patients (to control the disease-specificity) and 22 HS through real-time PCR. The microarray results showed higher levels of U13 small nucleolar RNA (SNORD13) in HD patients than controls (fold change 1.54, *p* = 0.003 HD vs. HS, and 1.44, *p* = 0.0026 HD vs. PP). In the validation population, a significant increase emerged with respect to both pre-HD and the control groups (*p* < 0.0001). SNORD13 correlated with the status of the mutant huntingtin carrier (r = 0.73; *p* < 0.001) and the disease duration (r = 0.59; *p* = 0.003). The receiver operating characteristic (ROC) curve analysis showed the high accuracy of SNORD13 in discriminating HD patients from other groups (AUC = 0.963). An interactome and pathway analysis on SNORD13 revealed enrichments for factors relevant to HD pathogenesis. We report the unprecedented finding of a potential disease-specific role of SNORD13 in HD. It seems to peripherally report a ‘tipping point’ in the pathogenic cascade at the neuronal level.

## 1. Introduction

Huntington’s disease (HD) is an inherited neurodegenerative disease caused by CAG trinucleotide repeat expansion in the first exon of the HTT gene, which encodes the huntingtin protein. HD is a progressive, incurable disease with a typical adult onset, related to CAG repeat length, and is characterized by motor impairment, cognitive dysfunction, and psychiatric symptoms. Disease-modifying treatments for HD are under development, and the identification of easily measurable biomarkers is crucial for predicting disease progression, monitoring the effects of novel drugs, and obtaining cues on the pathogenic cascade at the neuronal level.

Peripheral biomarkers are quantified in body fluids with minimal invasiveness, good accuracy, and a high discriminatory power. Cerebrospinal fluid (CSF) has been a focus of interest as a proxy for central nervous system (CNS) pathophysiology, and recent studies have identified reliable biomarkers, such as CSF mutant huntingtin (mHTT), used as an outcome measure for therapeutic approaches [1,2], which showed a very good predictive power for disease manifestation [3,4]. Biomarkers based on complex techniques or CSF may be of limited use because of their invasiveness, high cost, or the need for specialized personnel.

In recent times, increased focus has been given to the testing of more easily measurable biomarkers from peripheral leukocytes and plasma; these are cheaper, less invasive, and potentially more adept in obtaining longitudinal profiles. Among these, the measurement of leukocyte telomere length (LTL), which shortens remarkably in pre-symptomatic HD (PreHD) is a possible measure of time to clinical onset [5]. The histone variant pγ-H2AX is a component of the DNA damage responses in peripheral blood mononuclear cells (PBMCs) and changes in its levels proved to be an informative, potentially reversible biomarker in pre-HD [6]. In other studies, the PBMCs from patients with HD have been used to study mHTT and other gene expression profiles as predictors of disease progression [7,8]. Plasma neurofilament light protein (NfL) is reliable in monitoring disease progression, although it is less sensitive than CSF NfL [4]. Other studies have reported informative results on the plasma levels of oxidative stress markers, metabolic markers, and immune system products [9]. Recent studies on small non-coding RNAs in the plasma of patients with HD have led to several investigations of circulating micro-RNAs (miRNAs) [10,11,12]. Our recent study found that hsa-miR-323b-3p is upregulated in individuals with an mHTT mutation [13]. In this context, we obtained results on small nucleolar RNAs (snoRNAs), which have not been previously studied in HD, and thus, we considered snoRNAs as a possible peripheral biomarker of disease and elucidated their role in disease pathogenesis and progression.

SnoRNAs are a class of non-coding small guide RNAs, most of which direct the chemical modifications of other RNA substrates, including ribosomal RNAs (rRNAs) and spliceosomal RNAs. Moreover, some snoRNAs are involved in the regulation of alternative splicing and post-transcriptional modifications of mRNA [14]. Homo sapiens U13 snoRNA (SNORD13), 104 nucleotides long, is a member of the Box C/D family of small nucleolar ribonucleoproteins that can form base-pair interactions with the 3’ portion of 18S rRNA and is involved in the processing of this rRNA [15].

This study is aimed at reporting changes in circulating SNORD13 levels in people with prodromal and overt HD compared with several groups of controls in order to avoid confounders and to verify the disease specificity of our finding.

## 2. Results

We performed an exploratory microarray study of whole noncoding RNA expression profiles in the plasma of nine patients with HD (mean age of 48.25 ± 10.47; four males and five females), and 13 controls including eight healthy subjects (HS, mean age of 49.17 ± 11.79; two males and six females) and five psychiatric patients (PP, mean age of 50.25 ± 11.47; two males and three females) with schizophrenia or bipolar disorder. As these are the patients with HD that are often treated with psychotropic drugs, we included PP with similar treatment profiles as a control group in order to minimize the possible iatrogenic impact on the profile of the small non-coding RNAs. In particular, we selected patients treated with olanzapine, lithium, and valproate which were the treatments most frequently prescribed to patients affected by HD.

The microarray results indicated that SNORD13 levels were increased in the plasma of patients with HD compared to those in the HS and PP control groups (fold change, 1.54; *p* = 0.0003 HD vs. HS, and fold change, 1.44; *p* = 0.0026 HD vs. PP; Figure 1A). To validate this result, SNORD13 plasma levels were quantified using real-time PCR in five cohorts of subjects: 22 HS, 23 symptomatic patients with HD, 15 patients with pre-manifest HD (pre-HD), 24 PP, and 28 patients with Alzheimer’s disease (AD). The last group was considered as a control for the disease specificity of our findings. Demographic and clinical characteristics of each group are shown in Table 1. No significant relationship was observed between SNORD13 plasma levels and age/sex at blood sampling in any group, except for females in the HS group (*p* = 0.04). No relationship was observed between SNORD13 levels and CAG repeat length in the subjects with pre-HD or HD (Table 2). Our analysis showed a statistically significant (*p* < 0.0001) increase in the plasma levels of SNORD13 in patients with HD, which clearly segregated patients with overt disease (HD) from controls and pre-HD subjects. The changes in the plasma level of SNORD13 in symptomatic HD patients were highly significant compared to those of both the pre-HD and the three control groups (HS, PP, and AD; Figure 1B, Table 2).

A positive linear correlation was observed between circulating SNORD13 levels and disease duration in patients with HD (r = 0.589, *p* = 0.003) (Figure 2A). A significant relationship was also observed between plasma SNORD13 and the UHDRS clinical score in mHTT carriers (r = 0.732, *p* < 0.001, Figure 2B), whereas it was not found to be significant in only overt patients with HD (not shown). These linear relationships remained significant when considering age as a covariate (partial correlations, r = 0.563, *p* = 0.01, and r = 0.685, *p* < 0.001, respectively).

Next, we assessed the accuracy of plasma SNORD13 as a biomarker of overt HD through ROC curve analysis. In discriminating symptomatic HD patients from pre-symptomatic HTT mutation carriers, SNORD13 displayed an extremely high accuracy (AUC = 0.963, Figure 3A); setting the cut-off point of SNORD13 levels at 0.58 allowed the identification of patients with HD with 95.88% sensitivity and 86.7% specificity. Moreover, SNORD13 appeared to be of potential utility in distinguishing symptomatic HD patients from control groups (AD, PP, HS; AUC = 0.953; Figure 3B), as well as pre-HD among controls (AUC = 0.955; Figure 3C), and to a lesser extent, in identifying HTT mutation carriers (AUC: 0.811; Figure 3D).

Finally, to investigate the biological landscape of the action of SNORD13, we constructed an interactome. We retrieved information on U13 snoRNA interactions with proteins (including known RNA-binding proteins (RBPs) and transcription factors (TF)), other snoRNAs, miRNAs, and rRNAs from the databases RNAinter [16], snoDB [17], and RNAct [18]. Additionally, we mapped the intra-network protein-protein interactions through STRING [19]. The final network comprised 456 SNORD3-interacting nodes: 258 TFs, 86 RBPs, 91 proteins, one long noncoding RNA, three miRNAs, 13 snoRNAs, and the 18 s rRNA ribosomal subunit (Figure 4A and Appendix A). A pathway analysis revealed enrichment of processes involved in transcriptional regulation and RNA metabolism (Figure 4B–D), referring to molecules mostly located in the nucleus and involved in genomic organization (Figure 4E). Of interest in HD is the emergence of nerve growth factor (NGF)-stimulated transcription associated with SNORD13 activity, suggesting a direct implication in neurodegenerative processes, as well as the interaction with molecules involved in the DNA damage response that has already been implicated in disease pathogenesis and that are useful as peripheral biomarkers [6,7,20].

## 3. Discussion

Our study highlights an unprecedented finding of the potential role of snoRNAs in HD. The main results are the following: (i) the levels of circulating SNORD13 are significantly increased in the overt disease compared to the prodromal phase of HD; (ii) the levels of this snoRNA are comparable between HS and patients with a pre-HD status; (iii) this finding seems specific for HD, since three groups of control (HS, PP on drugs similar to that administered to patients with HD, and patients with AD) showed normal, comparable values; (iv) the plasma levels of SNORD13 seems to be related to the natural history of HD, correlating with the status of mHTT carriers and the disease duration; and (v) the above points (iii) and (iv) suggest that increased levels of SNORD13 in blood mirror pathogenic events in the CNS (though this fact awaits formal demonstration), possibly paving the way for new therapeutic targets.

A possible explanation for our data is that the elevated plasma level of SNORD13 in symptomatic HD patients may be due to nucleolar stress caused by the presence of mutant RNAs that carry an expanded CAG repeat (expanded CAG RNAs). In recent years, increasing attention has been directed toward understanding the pathogenic mechanisms of mHTT, and several studies have supported the hypothesis that expanded CAG RNAs induce apoptosis by activating the nucleolar stress pathway in both patients with HD and transgenic models of the disease [21,22]. Specifically, it has been shown that expanded CAG RNAs compete with nucleolin (a multifunctional protein that is mainly localized in the nucleolus and involved in various steps of ribosome biogenesis) for the rRNA promoter, leading to the reduction of rRNA expression, nucleolar stress and apoptosis via p53, and activation of the downstream signaling cascade, including mitochondrial cytochrome C release and caspase activation [23,24,25,26].

However, considering the SNORD13 interactome, we cannot rule out the possibility that other pathogenic mechanisms might be involved in the change in the levels of snoRNAs. Indeed, beyond ribosomal biogenesis and RNA metabolism, SNORD13 appears to be involved in a wide range of genomic activities associated with HD pathophysiology (Figure 4, Appendix A): chromatin remodeling via histone acetylation and methylation (SIRT6, EP300, TAF1, CHD1, KDM5A-B etc.), telomere length maintenance (DKC1,HRNPU, VPRBP) [5], DNA repair and damage response (ERCC6, BAZ1B, FANCD2, TOPBP1) [6,20]; direct modulation of homeobox (HOXA1-7, DLX1-2, OTX2 and others), and zinc-finger proteins (SNAI2, SP1-2, ZMIZ2 and others) [27]. We also identified three miRNAs (hsa-miR-455-5p, hsa-miR-342-3p, and hsa-miR-377-3p) which may cooperate with U13 snoRNA in regulating gene expression as being possibly affected in HD; stochastic computational analyses might help further explain these relationships, as piloted in multiple sclerosis pathogenesis modeling [28], COVID epidemic wave stability evaluation [29], and many other complex problems.

Irrespective of the mechanism(s) possibly linking mHTT to SNORD13 and the source of this snoRNA (currently unknown), we can speculate that the increase in the plasma level of SNORD13 in patients with HD may peripherally report a ‘tipping point’ in the pathogenic cascade at the neuronal level, while normal levels may mark a ‘molecular pre-manifest status’ in disease evolution. In fact, when we used published data [2] on CSF mHTT and plasma or CSF NfL as benchmarks to compare the performance of plasma SNORD13, we found that it outperformed both CSF mHTT and plasma or intrathecal NfL as a reporter of overt HD, supporting its potential value as a peripheral cue of central pathogenic processes (Table 3) In a clinical context circulating SNORD13 does not have a predictive value, being absent in pre-HD; however, this easily measurable, inexpensive and reliably quantifiable test (it is consistent, accurate, sensitive, specific, and reproducible) [9] may be useful to better catch, on a laboratory basis, the shift from prodromal to overt HD [30]. This may be useful to stratify and select candidate patients for clinical trials and to aim at disease-specific pathophysiological cascades.

In summary, circulating SNORD13 could be a clinically actionable asset in future trials as a peripheral reporter of a ‘functional reserve’ at the neuronal level, as well as a hint for new therapeutic targets.

Our research is subject to several limitations: the first is the small sample size, but we performed a sample size calculation to reduce this type of bias; the second limitation is the selection bias due to the different age of onset of each disease group taken into exam (AD patients are usually older than HD patients while PP are usually younger); and lastly, our results derive from a cross-sectional study and they consequentially reflect SNORD13 plasma levels at a single point in time both in HD, pre-HD and controls.

Further longitudinal studies with a larger sample size will be necessary to reduce these limitations and to validate our hypothesis.

In conclusion, peripheral small RNAs may offer potential advantages in terms of disease specificity compared to other approaches, such as circulating NfL, which has already been reported as a promising peripheral biomarker for many neurological diseases [3]. Future personalized therapies for different phases of the HD course and possible trials with etiologic approaches in subjects with pre-HD are necessary, which will plausibly introduce circulating biomarkers such as SNORD13 as promising components of the neurogeneticist’s toolkit.

## 4. Materials and Methods

### 4.1. Study Population

Participants in the study were enrolled at the Center for Experimental Neurological Therapies, Unit of Neurology (patients with a positive test for HD, patients with probable Alzheimer’s disease, and healthy subjects), and the School of Medicine and Psychology, Unit of Psychiatry (patients with psychiatric disorders), S. Andrea Hospital, Department of Neurosciences, Mental Health, and Sensory Organs, Sapienza University of Rome, Italy. The study was approved by the local ethics committee and written consent was obtained from all participants according to the principles of the Declaration of Helsinki. The operators were unaware of the disease state of each sample during processing and statistical analysis. The eligible subjects for this study were patients with a positive genetic test for HD, a diagnosis of schizophrenia or bipolar disorders treated with antipsychotic drugs, a diagnosis of probable Alzheimer’s disease, and healthy subjects (controls). Exclusion criteria were pregnancy, breastfeeding, and severe systemic illnesses or conditions.

### 4.2. Plasma Preparation and Affymetrix Gene Chip microRNA (miRNA) Array

Blood samples were obtained via venous puncture in ethylenediaminetetraacetic acid (EDTA) tubes for plasma preparation. The plasma was obtained by centrifugation (1500× *g* for 15 min at 4 °C); a few 500 μL aliquots of supernatant were stored at −80 °C. RNA was extracted using a Plasma/Serum Circulating RNA Purification Kit (NORGEN) following the manufacturer’s instructions. RNA quality and purity were assessed using an RNA 6000 Nano Assay kit on an Agilent 2100 Bioanalyzer (Agilent Technologies, Santa Clara, CA, USA). Briefly, 500 ng of total RNA was labeled using FlashTag Biotin HSR (Genisphere, Hatfield, PA, USA) and hybridized to GeneChipR miRNA 2.0 Arrays. The arrays were stained in Fluidics Station 450 and scanned using a GeneChip R Scanner 3000 (Affymetrix, Santa Clara, CA, USA). A statistical analysis was performed using Transcriptome Analysis Console (TAC) software (Thermo Fisher Scientific, Waltham, MA, USA). To survey the presence of outliers that could impact the dataset, principal component analysis (implemented in R) was performed to identify possible outliers that needed to be excluded. MiRNA probe outliers were defined according to the manufacturer’s instructions (Affymetrix), and further analysis included data summarization, normalization, and quality control using the web-based miRNA QC Tool software (Affymetrix). The raw dataset is available from the Gene Expression Omnibus (GEO) repository (GSE167630).

### 4.3. Real Time PCR Analysis

RNA extraction was performed from plasma using the miRNeasy Serum/Plasma kit according to the manufacturer’s protocol (Qiagen, Hilden, Germany). A U13 snoRNA analysis was performed by quantitative reverse transcription PCR (qRTPCR). cDNA was synthesized using the TaqMan™ MicroRNA Reverse Transcription Kit (Life Technologies, Carlsbad, CA, USA), according to the manufacturer’s instructions using a U13 snoRNA-specific primer (manufacturer-provided). Quantitative real-time PCR was performed on an ABI 7300 Real-time PCR System (using custom TaqMan^®^ Small RNA Assay (Life Technologies), according to the manufacturer’s instructions. Relative quantitation of U13 snoRNA was performed by the delta Ct method, using U6 snoRNA as an endogenous control [32]. Replicate assays of the same sample were performed to calculate inter-assay variation. The average standard deviation (SD) calculated by measuring the plasma SNORD13 levels of a sample repeated over three different assays was 0.035%. Thus, assuming a normal distribution, samples differing in average SNORD13 levels by as little as 0.069% (1.96 × SD) should be distinguishable by this method at a 95% confidence interval [33].

### 4.4. Statistical Analysis

Statistical analyses were performed using Partek Genomic Suite software (miRNA Array data), GraphPad Prism v9, and R (The R Project for Statistical Computing) v3.6.3. Data normality was assessed using the Shapiro–Wilk test. Nonparametric tests were used to compare the distribution of SNORD13 plasma levels between the patients and controls.

Nonparametric tests or linear regression were used to evaluate the distribution of SNORD13 plasma levels across age, sex, and CAG repeat number. Spearman’s correlation was computed to assess the linear relationship between SNORD13 plasma levels and disease duration or the Unified Huntington’s Disease Rating Scale-Total Motor Score (UHDRS-TMS), and age was considered a covariate in the partial correlations. Statistical significance was set at *p* < 0.05.

The sample size calculation was based on our preliminary study that enrolled 10 HD subjects and 10 controls, detecting a difference between group means of 0.58 with a standard deviation of 0.34. Assuming α = 0.01 and equal group size, we would need to study 12 HD and 12 control subjects to reject the null hypothesis that the population means of the experimental and control groups are equal with probability (power) 0.9.

A receiver operating characteristic (ROC) curve analysis was performed using the R easy ROC web interface. The optimal cutoff point was identified according to the Youden index method. Sample size estimation settings were as follows: type I error, 0.05; power, 0.8; allocation ratio, 1; area under the curve (AUC) derived from the HD/pre-HD ROC curve.

### 4.5. Interactome Construction and Analysis

The molecular interactions of SNORD13 were screened in three databases: (1) the RNA interactome database (RNAinter v4.0) [34] at http://www.rna-society.org/rnainter/ (accessed on 1 November 2021), which includes more than 41 million predicted or experimentally validated interactions; (2) snoDB, the largest repository of snoRNA biological annotation and manually curated snoRNA-RNA interactions [35]; (3) RNAct, a database of genome-wide predicted protein–RNAs interactions [17]. Nonhuman interactors were excluded from this study. Protein-protein interactions were derived by querying the STRING database [18] for high-confidence interactions (score > 0.7) of SNORD13-interacting proteins. The global SNORD13 interactome was mapped using Cytoscape [19]. A pathway enrichment analysis was performed using the STRING application in Cytoscape. Graphical plotting was performed using the ggplot2 package in R [20].

## Figures and Tables

**Figure 1 ijms-23-12440-f001:**
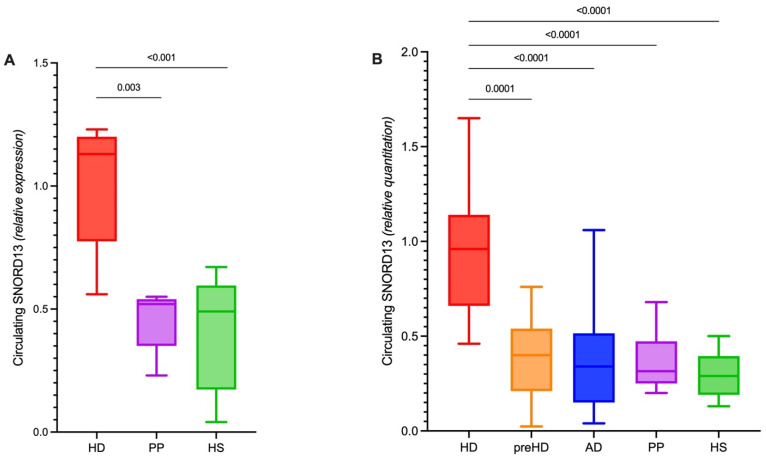
HD-specific signature of circulating SNORD13. (**A**) Relative expression of circulating SNORD13 measured through microarray in a cohort of HD patients, psychiatric patients (PP) and healthy subjects (Mann-Whitney test). (**B**) Relative quantitation of circulating SNORD13 measured trough RT-PCR in an independent cohort of HD patients, premanifest mHTT carriers (pre-HD), people with Alzheimer’s disease (AD), psychiatric patients (PP) and healthy subjects (HS) (Kruskal-Wallis test, Dunn’s test for multiple comparisons).

**Figure 2 ijms-23-12440-f002:**
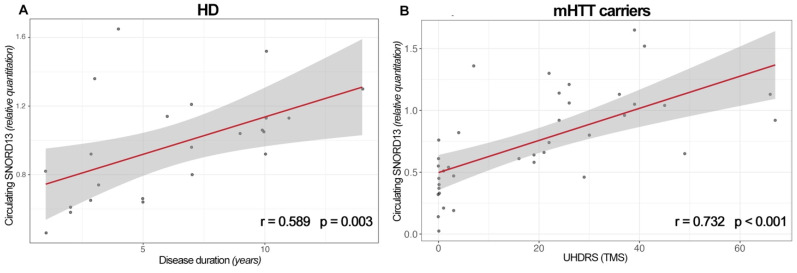
Correlation between SNORD13 plasma levels and (**A**) disease duration in HD patients, and (**B**) Unified Huntington’s disease Rating Scale Total Motor Score (UHDRS-TMS) in mHTT carriers (Spearman’s correlation).

**Figure 3 ijms-23-12440-f003:**
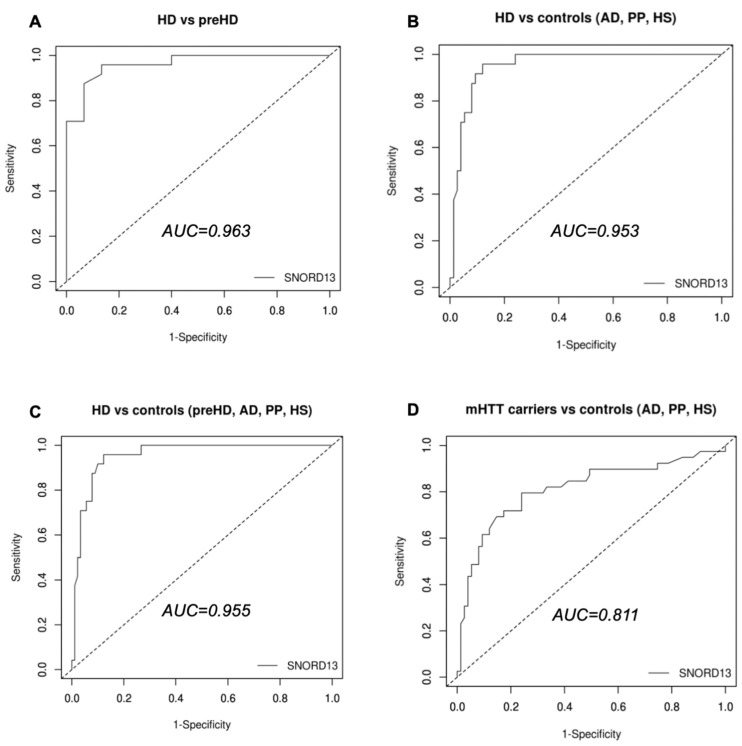
Assessment of SNORD13 as a biomarker of HD. ROC curves for discrimination between (**A**) symptomatic HD and premanifestomatic HD (pre-HD); (**B**,**C**) symptomatic HD and control groups; (**D**) mHTT carriers (HD and pre-HD) and control groups.

**Figure 4 ijms-23-12440-f004:**
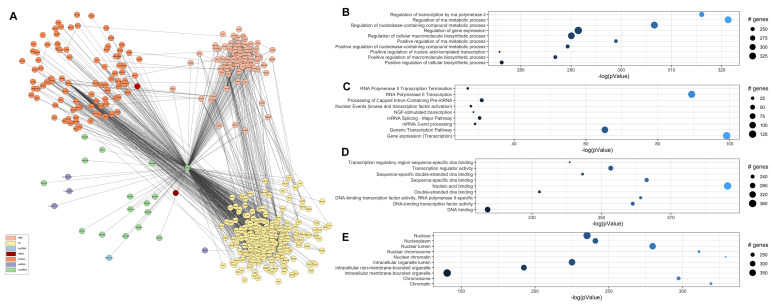
SNORD13 interactome and pathway analysis. (**A**) SNORD13 interaction map. Nodes are colored according to biological functions (see legend). (**B**–**E**) Top 10 results of enrichment analysis of SNORD13 interactors in: (**B**) Gene Ontology Biological Processes; (**C**) Reactome pathways; (**D**) Gene Ontology Molecular Function; (**E**) Gene Ontology Cellular Compartment. *X*−axis and color represent the enrichment strength in terms of –log (*p*-value); dot size is proportional to the number of genes enriched in each set.

**Table 1 ijms-23-12440-t001:** Demographic and clinical characteristics of mHTT carriers and controls. UHDRS-TMS: Unified Huntington’s Disease Rating Scale Total Motor Score; TFC: Total Functional Capacity; SDMT: Symbol Digit Modality Test.

	Healthy Subjects *n* = 22	Pre-Manifest mHTT Carriers *n* = 15	HD Patients *n* = 23	Psychiatric Patients *n* = 24	AD Patients *n* = 28
Age (years)	50.3 ± 13.4 (range 26–76)	36.4 ± 12.4 (range 21–59)	53.2 ± 11.4 (range 28–75)	48.1 ± 13.1 (range 20–67)	67.6 ± 5.3 (range 56–76)
Sex (males, %)	40.9	26.6	69.6	66.7	63
CAG Median (range)	/	42.8 (40–49)	42.8 (40–47)	/	/
UHDRS-TMS	/	1.1 ± 2.1	30.2 ± 16.0	/	/
TFC	/	13	9.1 ± 2.7	/	/
SDMT	/	55.8 ±17.7	18.6± 10.2	/	/
Disease duration	/	/	6.2 ± 3.7 years (range 1–14)	/	3.8 ± 1.7 years (range 1–7)

**Table 2 ijms-23-12440-t002:** Circulating SNORD13 in HD and control groups (referring to Figure 1B).

Group (*n*)	SNORD13 Relative Quantitation (Median, 1st and 3rd Quartiles)	*p*-Value vs. HS	Covariate Analysis		
			SNORD13/Age (*p*-Value)	SNORD13/Sex (*p*-Value)	SNORD13/CAG Number (*p*-Value)
HS (22)	0.29 (0.19–0.41)	/	0.21	0.04	
pre-HD (15)	0.40 (0.21–0.54)	0.1	0.48	0.44	0.37
HD (23)	0.96 (0.66–1.14)	<0.0001	0.53	0.33	0.77
PP (24)	0.32 (0.25–0.47)	0.24	0.99	0.9	/
AD (28)	0.34 (0.15–0.52)	0.55	0.19	0.84	/

**Table 3 ijms-23-12440-t003:** Diagnostic ability of SNORD13 compared with NFL and mHTT performance from the work by Byrne et al., 2018 [31] (AUC derived from ROC curve analysis).

	Controls vs. mHTT Carriers	Pre-HD vs. HD
CSF mHTT	1.000	0.778
CSF NfL	0.933	0.914
Plasma NfL	0.914	0.931
Plasma SNORD13	0.811	0.963

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
