# Peer review of "Circulating U13 Small Nucleolar RNA as a Potential Biomarker in Huntington’s Disease: A Pilot Study"

_ijms, 2022, doi:10.3390/ijms232012440_

Round 1

Reviewer 1 Report

The authors investigated the diagnostic accuracy of SNORD13) as a biomarker in Huntington's disease (HD) in this study. Although the study has scientific merit, I have several concerns those needs to be addressed, which are as follows:

1-     Introduction: It is critical to express the rationale for SNORD13 selection in this study in terms of its function, as well as its relationship to HD and pathogenesis.

2-     Materials and methods:

-        2.3. Real-Time PCR analysis: Include the primer sequences for U13 snoRNA and U6 snoRNA.

-        2.4. Statistical Analysis: The sample size was determined. However, the required sample size for this study for each group is not provided.

3-     Results: Other markers for HD, such as CSF mHTT, CSF, and plasma NfL, should be assessed in the studied groups and their accuracy compared to SNORD13, as retrieving their results from a previous study appears to be illogical.

4-     Discussion: Limitations such as small sample size, lead-time bias, and selection bias should be acknowledged.

5-     Abbreviations such as NfL, UHDRS-TMS, TFC, and SDMT should be defined when they are first used in the text.

6-     Minor English editing is required.

Author Response

1. Introduction: It is critical to express the rationale for SNORD13 selection in this study in terms of its function, as well as its relationship to HD and pathogenesis.

Answer: We selected the U13 snoRNA through an exploratory microarray study of whole noncoding RNA expression profiles in the plasma of patients with HD. As this snoRNA has not been previously studied neither in HD nor in other neurodegenerative diseases, we formulated a hypothesis about its possible role in the pathogenesis of HD, and as such, we have included it in the discussion.

2. Materials and methods:

 2.3. Real-Time PCR analysis: Include the primer sequences for U13 snoRNA and U6 snoRNA.

Answer: The primers were provided by the manufacturer (included in the TaqMan® Small RNA Assays) who did not give us the sequences.

 2.4. Statistical Analysis: The sample size was determined. However, the required sample size for this study for each group is not provided.

Answer: We thank the reviewer for this suggestion and we added information about the required sample size in the text at pag 9:

Sample size calculation was based on our preliminary study that enrolled 10 HD subjects and 10 controls, detecting a difference between group means of 0.58 with a standard deviation of 0.34. Assuming α=0.01, equal group size, we would need to study 12 HD and 12 control subjects to reject the null hypothesis that the population means of the experimental and control groups are equal with probability (power) 0.9.

 3.  Results: Other markers for HD, such as CSF mHTT, CSF, and plasma NfL, should be assessed in the studied groups and their accuracy compared to SNORD13, as retrieving their results from a previous study appears to be illogical.

Answer: In agreement with the reviewer’s suggestion we have eliminated from the results the comparison between SNORD13 and the other HD biomarkers, furthermore we have moved the Tab3 in the discussion (or in the supplementary).

4.  Discussion: Limitations such as small sample size, lead-time bias, and selection bias should be acknowledged.

Answer: We added in the discussion more information about study limitations (pg 8), as suggested by the reviewer:

However, our research is subject to several limitations: the first is the small sample size, however we performed a sample size calculation to reduce this type of bias; the second limitation is the selection bias due to the different age of onset of each disease taken into exam (AD patients are usually older than HD patients while PP are usually younger); at last, our results derive from a cross-sectional study and they consequentially reflect SNORD13 plasma levels at a single point in time both in HD, pre-HD and controls. Further longitudinal studies with a larger sample size will be necessary to validate our hypothesis.

5. Abbreviations such as NfL, UHDRS-TMS, TFC, and SDMT should be defined when they are first used in the text.

Answer: According to reviewer suggestion we have defined the acronyms in the text.

6.  Minor English editing is required.

Answer: In agreement with reviewer’s suggestion the manuscript was reviewed by a native English speaker, using an English editing service (Wiley Editing Services).

Reviewer 2 Report

The manuscript is highly interesting. The authors have made an extraordinary effort in describing the problem and guiding the readers through their experiments, which demonstrate that SNORD13 can be used as a biomarker for Huntington's disease, with the advantage that it can be measured in the patient's plasma. A very nice piece of work.

Author Response

We thank the reviewer for the positive comments on our work. 

Round 2

Reviewer 1 Report

The authors have adequately addressed all my concerns and queries.